# Interferon-Induced Protein 6-16 (IFI6-16) from *Litopenaeus vannamei* Regulate Antiviral Immunity via Apoptosis-Related Genes

**DOI:** 10.3390/v14051062

**Published:** 2022-05-16

**Authors:** Kai Lǚ, Haoyang Li, Sheng Wang, Anxing Li, Shaoping Weng, Jianguo He, Chaozheng Li

**Affiliations:** 1State Key Laboratory of Biocontrol, Southern Marine Science and Engineering Guangdong Laboratory (Zhuhai), School of Marine Sciences, Sun Yat-sen University, Guangzhou 510275, China; lvkai3@mail2.sysu.edu.cn (K.L.); lihy88@mail.sysu.edu.cn (H.L.); wangsh336@mail.sysu.edu.cn (S.W.); lianxing@mail.sysu.edu.cn (A.L.); lsswsp@mail.sysu.edu.cn (S.W.); 2Guangdong Provincial Key Laboratory of Marine Resources and Coastal Engineering/Guangdong Provincial Key Laboratory for Aquatic Economic Animals, School of Life Sciences, Sun Yat-sen University, Guangzhou 510275, China; 3Maoming Branch, Guangdong Laboratory for Lingnan Modern Agriculture, Maoming 525000, China

**Keywords:** IFN stimulated gene, IFI6-16, JAK/STAT pathway, caspase, *Litopenaeus vannamei*, WSSV

## Abstract

A growing number of evidence shows that some invertebrates possess an antiviral immunity parallel to the interferon (IFN) system of higher vertebrates. For example, the IRF (interferon regulatory factor)–Vago–JAK/STAT regulatory axis in an arthropod, shrimp *Litopenaeus vannamei* (whiteleg shrimp) is functionally similar to the IRF–IFN–JAK/STAT axis of mammals. IFNs perform their cellular immunity by regulating the expression of target genes collectively referred to as IFN-stimulated genes (ISGs). However, the function of invertebrate ISGs in immune responses is almost completely unclear. In this study, a potential ISG gene homologous to the interferon-induced protein 6-16 (IFI6-16) was cloned and identified from *L. vannamei*, designated as LvIFI6-16. LvIFI6-16 contained a putative signal peptide in the N-terminal, and a classic IFI6-16-superfamily domain in the C-terminal that showed high conservation to other homologs in various species. The mRNA levels of LvIFI6-16 were significantly upregulated after the stimulation of poly (I:C) and challenges of white spot syndrome virus (WSSV). Moreover, silencing of LvIFI6-16 caused a higher mortality rate and heightened virus loads, suggesting that LvIFI6-16 could play a crucial role in defense against WSSV. Interestingly, we found that the transcription levels of several caspases were regulated by LvIFI6-16; meanwhile, the transcription level of LvIFI6-16 self was regulated by the JAK/STAT cascade, suggesting there could be a JAK/STAT–IFI6-16–caspase regulatory axis in shrimp. Taken together, we identified a crustacean IFI6-16 gene (LvIFI6-16) for the first time, and provided evidence that the IFI6-16 participated in antiviral immunity in shrimp.

## 1. Introduction

The interferon (IFN) can stimulate a robust innate immune response that plays a key role in host defense against invading microbes [1]. Following pathogen detection and subsequent IFN production, IFNs bind to cell surface receptors and launch a signaling cascade through the Janus kinase signal transducer and activator of transcription (JAK-STAT) pathway, resulting to the transcriptional regulation of hundreds of IFN-stimulated genes (ISGs) [2]. Induction of ISGs establishes a remarkable antiviral state, effective against RNA viruses, DNA viruses, and intracellular bacteria and parasites [1]. IFN signaling also plays a key role in shaping the adaptive immune response [3]. Studies of ISGs action have yielded important findings including genetic modification, gene regulation, protein transport and degeneration, and other cellular processes [1]. IFI6-16 and ISG15 were among the first ISGs identified [4], and ISG12 was then identified as an ISG related to IFI6-16 [5]. Although robust induction of IFI6-16 by IFNs implies its critical functions in innate immunity, it has been less comprehensively characterized than some other ISGs, especially in invertebrates. 

As one of the innate immune effectors in the IFN antiviral pathway, IFI6-16 participates in apoptosis response during virus infection in vertebrates [5,6]. IFI6-16 is an apoptosis-related protein in the early period of virus invasion [7], and has a high expression profile in some tumor tissues [8,9]. Shrimp *Litopenaeus vannamei* is the uppermost shrimp species that accounts for 80% of shrimp production worldwide, and thus has received a great deal of attention to understand the relationship between host immunity and diseases outbreak [10]. Upon the threat of diverse pathogens, such as *Vibrio parahaemolyticus (V. parahaemolyticus)*, white spot syndrome virus (WSSV) and Decapod iridovirus 1 (DIV1), several distinct antimicrobial immune pathways have been discovered in shrimp *L. vannamei* [10,11,12,13]. Shrimp diseases have led to huge economic losses, especially the white spot disease (WSS), which caused by WSSV. Annual losses ($1 billion) caused by WSS have traditionally equated to ~10% of global shrimp production. There is still a debate about whether there is an IFN antiviral pathway in invertebrates. In recent years, a prominent finding is that the IRF–Vago–JAK/STAT regulatory axis in shrimp could be functionally equivalent to IRF–IFN–JAK/STAT regulatory axis in higher vertebrate cells [14]. ISGs regulated by JAK/STAT cascade is the antiviral executor or effector molecules that perform antiviral function via enhancing host resistance such as inducing apoptosis [5], as well as acting on virus such as disturbing virus replication [15]. However, the information about the function of invertebrate ISGs is not understood. 

In the present study, we cloned and identified an IFI6-16 homolog from *L. vannamei* named LvIFI6-16, which showed an evolutionary conservation in the IFI6-16 superfamily domain. LvIFI6-16 was significantly upregulated after WSSV infection, and performed an antiviral function probably via regulating the expression of apoptosis-related genes. These findings will contribute to a better understand of the interferon system-like antiviral mechanism in invertebrates, as well as provide some insights into the resistant breeding for disease control in shrimp aquaculture. 

## 2. Materials and Methods

### 2.1. Cloning of Full Length of LvIFI6-16 cDNA

Based on *L. vannamei* EST and genome data [16], a putative IFI6-16 protein was retrieved to clone the full length of LvIFI6-16 with the gene-specific primers (Table 1) through the rapid amplification cDNA ends (RACE) method. The cDNA library for RACE PCR was constructed with the SMARTer PCR cDNA Synthesis Kit (Clontech) according to the user’s instructions. The first round of PCR amplifications was conducted on 20-fold dilutions of SMART RACE cDNA with Universal Primer A Mix (UPM)/LvIFI6-16-5’RACE1 (for 5’-RACE) or UPM/LvIFI6-16-3’RACE1 (for 3’-RACE), respectively. Samples were firstly denatured at 95 ℃ for 3 min followed by 5 cycles of 10 s at 94 ℃ and 1 min at 72 ℃; and then 30 cycles of 10 s at 94 ℃, 20 s at 62 ℃, and 1 min at 72 ℃; and a final 5 min extension at 72 ℃. The products of the first round RACE PCR were 20-fold diluted as templates for the second round RACE PCR. Primers of Nested Universal Primer A (NUP) and LvIFI6-16-5’RACE2 or 3’RACE2 were used for the second round RACE PCR. The procedures of the second round RACE PCR were the same as those of the first round. The second round RACE PCR products were purified and cloned into pEASY-T1 Cloning Vector (TransGen Biotech, Guangzhou, China) for sequencing, and finally 16 positive clones were selected to sequence.

### 2.2. Sequence and Phylogenetic Analysis of LvIFI6-16

Protein domains of LvIFI6-16 were identified by BLASTP and SMART (Simple Modular Architecture Research Tool, http://smart.embl.de/, accessed on 5 April 2021) [17]. Protein sequences of IFI6-16 homologs were retrieved from NCBI database by BLAST. We aligned LvIF16-16 and its homologs with the Clustal X 2.0 program [18], and analyzed the identities and similarities with GeneDoc software (http://www.nrbsc.org/gfx/genedoc/, accessed on 10 April 2021). The phylogenetic tree was generated based on the full-length amino acid sequences of IFI6-16 proteins by MEGA 5.0 software [19] with the neighbor-joining (NJ) method. 

### 2.3. Confocal Laser Scanning Microscopy

The open reading frame (ORF) without a stop codon of LvIFI6-16 was cloned into pAc5.1-GFP (Invitrogen, Waltham, MA, USA) vector at KpnI and EcoRI sites to generate pAc-LvIFI6-16-GFP plasmid for expressing GFP-tagged LvIFI6-16 protein. *Drosophila* S2 cells were seeded onto glass cover slips in a 12-well plate with approximately 40% confluent. Each well was transfected with 1 μg pAc5.1-LvIFI6-16-GFP plasmid using the FuGENE HD Transfection Reagent (Promega, Madison, WA, USA). At 48 h post transfection, subcellular localization of LvIFI6-16 was carried out by the Hoechst Staining Kit (Beyotime, Guangzhou, China). GFP monoclonal antibody (SAB2702197, Sigma-Aldrich, St Louis, MO, USA) and β-actin monoclonal antibody (A1978, Sigma-Aldrich, St Louis, MO, USA) were used to mark the LvIFI6-16-GFP protein and the cytoskeleton, separately. Briefly, cells were fixed, permeabilized, and incubated with primary antibodies, followed by incubation with secondary antibodies. Whereafter, the cells were stained with 2 mg/mL Hoechst 33258 (Beyotime, Guangzhou, China) for 5 min and washed three times with PBS, and finally visualized with confocal laser scanning microscope (Leica TCS-SP5, Heidelberg, Germany). 

### 2.4. The Quantitative RT-PCR Analysis of LvIFI6-16 Expression 

Disease-free shrimps (*L. vannamei*, ~5 g weight each) were obtained from the local shrimp farm in Maoming, Guangdong Province, China. For the tissue distribution assay, we collected 12 different shrimp tissues including muscle, eyestalk, scape, epithelium, hemocyte, gill, hepatopancreases, intestine, stomach, heart, nerve and pyloric caecum from 30 shrimps. For challenge experiments, 600 shrimps were divided into six groups (100 shrimps each group), in which each shrimp was injected with approximate 1 × 10^5^ CFU of *Staphylococcus aureus*, approximate 1 × 10^5^ CFU of *V. parahaemolyticus* or approximate 1 × 10^5^ copies of WSSV particles in 50 µL PBS, 5 µg LPS (Sigma-Aldrich, St Louis, MO, USA), 5 µg Poly (I:C) (Sigma-Aldrich, St Louis, MO, USA), respectively. The negative control group injected 50 µL PBS only. Shrimps were injected by using 1 mL insulin syringes on the second abdominal segment. Gills of challenged shrimps were collected at 0, 4, 8, 12, 24, 36, 48, and 72 h post injection, and the samples were pooled from 10 shrimps each time point. Total RNA was isolated by Trizol reagent (Life Technologies, Gaithersburg, MD, USA), and dissolved in nuclease free-water (Takara, Dalian, China). Total RNA (1 µg) was used in 20 µL of reverse transcription reaction by using TransScrip One-Step gDNA Removal and cDNA Synthesis SuperMix for the PCR kit (TransGen Biotech, Guangzhou, China) for the synthesis of the first-strand cDNA. Expression levels of LvIFI6-16 were monitored by quantitative PCR using SYBR Green Master Mix (Takara, Dalian, China) and calculated by employing the Livak (−2^△△CT^) method [20] after normalization to *L. vannamei* EF-1α. Primer sequences were listed in Table 1. All samples were tested in triplicate. 

### 2.5. Detection of Expression Levels of LvIFI6-16 and Apoptosis-Associated Genes in LvIFI6-16 Silenced L. vannamei 

The primers with a 5’ T7 RNA polymerase binding site (Table 1) were used to construct GFP, LvIFI6-16, LvIRF, and LvSTAT dsRNA by using T7 RiboMAX Express RNAi System Kit (Promega, Madison, WA, USA). The experimental groups were treated with injections of LvIFI6-16, LvIRF, or LvSTAT dsRNA, while the control groups were injected with GFP dsRNA (each shrimp was injected with 10 µg dsRNA in 50 µL PBS). Real-time quantitative RT-PCR was performed to detect the knockdown efficiency. Briefly, at 48 h after the dsRNA injection, gills of each group (10 shrimps) were collected for total RNA extraction and the synthesis of the first-strand cDNA as the template for quantitative PCR. Meanwhile, the expression levels of IFI1-16, IRF, JAK, STAT, and several apoptosis-associated genes including cytochrome c (CYC), apoptosis-inducing factor (AIF), and Caspase1-5 in gills of LvIFI6-16 knockdown shrimps at 48 h were also detected. *L. vannamei* EF-1α was used here as an internal control. Primer sequences were listed in Table 1.

### 2.6. WSSV and PBS Challenge Experiments in LvIFI6-16-Knockdown Shrimp

Disease-free shrimps (average 5 g, *n* = 30 each group) were injected with 50 µL dsRNA solution (LvIFI6-16 dsRNA or GFP dsRNA diluted in PBS) at a dose of 10 µg dsRNA per shrimp or PBS only. Forty-eight hours later, shrimps were injected with approximately 1 × 10^5^ copies of WSSV particles, and mock-challenged with PBS as a control, respectively. Shrimps were cultured in an aquarium with air-pumped circulating seawater and fed with an artificial diet four times a day for about 7 days following infection. The mortality of each group was observed every 4 h. The log-rank (Mantel-Cox) test was used to analyze differences between groups with the GraphPad Prism software. 

For the WSSV challenged groups, a collateral experiment was performed to monitor the WSSV replication in LvIFI6-16-knockdown shrimps (*n* = 30 for each group). Briefly, three shrimp gill tissues were mixed together as one sample from each group at 48 h post-WSSV infection. Gill DNA was extracted with TIANamp Marine Animals DNA Kit (TIANGEN, Guangzhou, China). The quantities of WSSV genome copies were monitored by using absolute quantitative PCR with primers WSSV32678-F/WSSV32753-R and a Taq Man fluorogenic probe WSSV32706 (Table 1) as described previously [21]. The WSSV genome copy numbers in 1 µg of shrimp gill DNA were then calculated. 

## 3. Results

### 3.1. Sequence Features of LvIFI6-16 

The full-length cDNA sequence of LvIFI6-16 was 801 bp, which contained a 291 bp 5’-untranslated region (UTR), a 99 bp 3’- UTR with a poly (A) tail, and a 411 bp open reading frame (ORF) coding for a putative 136-amino-acid protein with a calculated molecular weight of approximate 13.0 kDa and a theoretical isoelectric point (pI) of approximate 8.8 (GenBank accession number: ON186541) (Figure 1a). Domain prediction analysis showed that LvIFI6-16 protein contained a signal peptide in N-terminal, followed by a transmembrane region (TM) at 35-57 aa and a conserved IFI6-16-superfamily domain of 57 amino acids at its C-terminal (Figure 1b).

### 3.2. Phylogenetic Analysis 

The neighbor-joining (NJ) phylogenetic tree revealed that LvIFI6-16 was separated as a solitary group, two homologs from chicken (*Gallus gallus*) and zebra finch (*Taeniopygia guttata*) were clustered together in an out group, as well as the homologs from other vertebrates were clustered together in one group within three sub-branches (Figure 1d). In addition, multiple sequence alignment showed that LvIFI6-16 shared moderate similarities with other IFI6 proteins (Figure 1c). These results demonstrated that LvIFI6-16 was a member of IFI6-16 superfamily. 

### 3.3. Expression Levels of LvIFI6-16 mRNA in Healthy and Immune-Challenged Shrimps

Quantitative RT-PCR showed that the mRNA of LvIFI6-16 was widely expressed in all examined tissues (Figure 2a). Tissue distribution revealed that the expression of LvIFI6-16 was highest in the intestine, moderate in hepatopancreas, stomach, heart, gill, and low in the epithelium, pyloric caecum, hemocyte, nerve, muscle, scape, and eyestalk (Figure 2a). Quantitative RT-PCR was executed to investigate the time-course expression changes of LvIFI6-16 in gills after immune challenges as well. In *L. vannamei* gills, the expression of LvIFI6-16 was downregulated during 4 to 48 h, and surprisingly upregulated with a 1.72-fold increase at 72 h after the LPS challenge (Figure 2b). In response to the *S. aureus* challenge, the expression level of LvIFI6-16 was dramatically down-regulated at 4 h with 0.25 times of that at 0 h, and continued to fall to 0.15 times of initial level at 12 h, and then ascended gradually at 24–72 h (Figure 2c). After *V. parahaemolyticus* challenge, LvIFI6-16 was remarkably downregulated at 4, 8 and 72 h, and maintained low expression levels at 12–48 h (Figure 2d). Upon WSSV challenge, LvIFI6-16 expression slightly declined at 4 h, but abruptly increased from 8 h and reached a peak of 434-fold increase at 72 h (Figure 2e). During the Poly (I:C) challenge, the LvIFI6-16 expression slightly increased at 4 h, but markedly declined at 8, 12 and 48 h, and finally dramatically rose to 5.50-fold of the initial level at 72 h (Figure 2f). Taken together, these results demonstrated that the expression of LvIFI6-16 was suppressed at the early stage of *V. parahaemolyticus*, *S. aureus*, LPS, and Poly (I:C) challenges, but strongly induced during WSSV infection. 

### 3.4. Subcellular Localization of LvIFI6-16 

The pAc-GFP plasmid was constructed to express a green fluorescent protein in S2 cells and showed ubiquitously distribution in the cytoplasm and the nucleus (as a control) (Figure 3). The GFP-tagged LvIFI6-16 plasmid was obtained by inserting the full-length ORF of LvIFI6-16 into the pAc-GFP plasmid. The GFP-tagged LvIFI6-16 protein was visualized in plasmid transfected S2 cells by a confocal laser scanning microscope. LvIFI6-16 was dispersedly presented in the cytoplasm but rarely in the nucleus, suggesting that LvIFI6-16 was a cytoplasm localized protein (Figure 3).

### 3.5. LvIFI6-16 Played a Key Role in Defense against WSSV 

The silencing efficiency of LvIFI6-16 was checked by quantitative RT-PCR. At 48 h after the injection of dsRNA, the transcription level of LvIFI6-16 was remarkably downregulated in LvIFI6-16 dsRNA injected group, while there was no suppressive effect on LvIFI6-16 in the GFP dsRNA injected group (Figure 4a). Shrimps were challenged with WSSV or PBS at 48 h post dsRNA injection and the mortality rates were recorded for a period of 144 h every 4 h after the challenge. At 64 h post-WSSV challenge, the cumulative mortality of the LvIFI6-16 dsRNA group was significantly higher than that of the GFP dsRNA group, and this situation lasted until 120 h (Figure 4b). To further evaluate the effect of LvIF6-16-knockdown in shrimps on WSSV replication, the virus copies of WSSV in gills were detected using absolute quantitative PCR. The WSSV genome copies of the dsLvIFI6-16 group were much higher than those of the dsGFP control group at 48 h post WSSV injection (Figure 4c). These results suggested that LvIFI6-16 could play a key role in antiviral defense, at least against WSSV. 

### 3.6. LvIFI6-16 Regulated Apoptosis-Related Genes with or without WSSV Challenge 

In order to explore whether LvIFI6-16 is regulated by IRF–JAK/STAT signaling pathway, we knocked down LvIRF or LvSTAT in shrimp for testing the mRNA level of LvIFI6-16. As shown in Figure 5a,b, the transcription levels of LvIFI6-16 in gills were obviously downregulated in LvIRF or LvSTAT silenced shrimps, which indicates that LvIFI6-16 could be positively regulated by LvIRF and LvSTAT. Further, mammalian IFI6-16 protein has been shown to promote cell apoptosis which is related to an antiviral response, so we sought to find whether LvIFI6-16 can regulate apoptosis-related genes. We observed that LvIFI6-16 could regulate several apoptosis-related genes with or without WSSV challenge. Specifically, the expression levels of LvCYC, LvAIF, LvCaspase1, LvCaspase3, and LvCaspase4 were obviously downregulated in LvIFI6-16 silenced shrimps (Figure 5c), while the LvCYC, LvAIF, LvCaspase1, LvCaspase2, LvCaspase3, and LvCaspase4 were also suppressed after WSSV infection (Figure 5d), which suggested that LvIFI6-16 played a positive role in regulating these genes expression. In addition, considering that these apoptosis-related genes usually execute pro-apoptosis function, we thus infer that LvIFI6-16 could perform its defense role via inducing antiviral apoptosis against WSSV. Furthermore, since many ISGs can feedback regulate some key molecules in their upstream signaling pathways [7], we thus detected whether LvIFI6-16 had an impact on the JAK/STAT signaling axis under WSSV infection. The results showed that silencing of LvIFI6-16 resulted in much lower expression levels of LvJAK and LvSTAT compared to the control group under WSSV stimulation (Figure 5e). Taken together, these results suggested there could be an IRF–JAK/STAT–IFI6-16–caspase regulatory axis in shrimp that conferred antiviral protection via inducing apoptosis, as well as that LvIFI6-16 could positively feedback regulate the JAK/STAT cascade that could establish a feedback-loop to enhance the activity of the whole regulatory axis (Figure 5f). 

## 4. Discussion

Until now, the mammalian IFN mediated signaling and antiviral mechanism has been well established since the phenomenon of ‘virus interference’ was discovered in 1957 [2]. In response to viral infection, cells produce and secrete a small protein (IFN) that induces to establish an antiviral state to clear infection or guard against invasion. Briefly speaking, the secreted IFN binds to its cognate cell-surface receptors to activate the JAK/STAT cascade that leads to hundreds of ISGs expression [22]. In recent years, a growing number of studies have shown that some invertebrates such as shrimps and oysters possess an IFN-like signaling pathway [14,23]. Nevertheless, this very limited information restricts our ability to understand such exciting IFN-like signaling in invertebrates. In this study, we cloned and identified an IFN-stimulated gene from shrimp *L. vannamei* (termed LvIFI6-16) for the first time. The LvIFI6-16 mRNA expression profiles in different tissues and in response to several immune stimuli were identified. We demonstrated that LvIFI6-16 conferred viral resistance to shrimp that could attribute to its ability to regulate apoptosis-related genes. 

LvIFI6-16 contained a putative signal peptide in N-terminal followed by a transmembrane region which indicated it could be an intracellular transmembrane protein. Generally speaking, the intracellular transmembrane proteins are mainly synthesized in the endoplasmic reticulum and transported to special organelles guided by some characteristic signal sequence [24], which could explain the ectopic expression of LvIFI6-16 was primarily located in the cytoplasm of *Drosophila* S2 cells. The exact subcellular localization of LvIFI6-16 needs to be addressed in the future. LvIFI6-16 also contained a conserved IFI6-16-superfamily domain at its C-terminal, suggesting that LvIFI6-16 has a similar structural feature to other IFI6-16 homologs. In the phylogenetic analysis of IFI6-16 homologs, LvIFI6-16 was assigned to a separate branch with only one member of itself, indicating that LvIFI6-16 was a novel member of the IFI6-16 family probably harboring some different functions. 

LvIFI6-16 was highly expressed in intestine, moderate in hepatopancreas, stomach, heart and gill, while these tissues are generally considered as the immune organs of shrimp. The extraordinary high expression level of LvIFI6-16 in intestine may imply its correlation with intestine health. As an important immune organ, the gill was chosen for explore the expression of LvIFI6-16 in response to different challenges including LPS, *V. parahaemolyticus*, *S. aureus*, Poly (I:C) and WSSV. Except for WSSV, these four stimuli resulted in an obviously low expression of LvIFI6-16 during early challenges (~48 h), which suggested that LvIFI6-16 could be a negative regulatory factor in response to these four stimuli. However, we did not know whether the low expression of LvIFI6-16 is an active or passive contribution by these four stimuli. Notably, LvIFI6-16 expression was dramatically upregulated during WSSV infection, even reaching a peak with a 434-fold increase at 72 h. We thus inferred that LvIFI6-16 could play a crucial role in host immune response to WSSV infection. This is confirmed by an in vivo experiment that knockdown of LvIFI6-16 elevated shrimp susceptibility to WSSV, as observed higher cumulative mortality and increased virus loads in LvIFI6-16 silenced shrimps. In vertebrates, IFI6-16 have also been shown to possess antiviral ability against different virus invasion. For example, overexpression of IFI6-16 inhibited genome replication and gene expression of HBV (hepatitis B virus) in HepG2 cells [25], and researchers also found that IFI6-16 could inhibit HCV (hepatitis C virus) infection by impairing EGFR mediated CD81 co-localization with claudin-1 [26]. Distinct IFI6-16 homologs from various species have some similarities and differences in the antiviral actions through which they exert their biological effects. One of the antiviral actions is that IFI6-16 was involved in regulating apoptosis with the ability to affect mitochondrial membrane potential during viral infection [6]. Here, we found that LvIFI6-16 was able to positively regulate the expression of several apoptosis-related genes including LvCYC, LvAIF, and LvCaspase1-5 that exhibited antiviral activity by inducing apoptosis [27,28,29]. We thus supposed that LvIFI6-16 elicited viral resistance to the host (shrimp) via their ability to promote apoptosis. Besides, LvIRF and LvSTAT were shown to participate in regulating the expression of LvIFI6-16, while LvIFI6-16 itself could feedback induce the expression of LvSTAT, suggesting there could be an IRF–JAK/STAT–IFI6-16 regulatory axis in shrimp. In vertebrates, JAK/STAT cascade is responsible for inducing ISGs, while some ISGs can feedback affect the activity of the JAK/STAT axis to establish a regulator loop [1,2]. A similar result was observed in the action of LvIFI6-16, which supported that LvIFI6-16 could be a typical ISG in shrimp based on both its antiviral function and its involving regulator cascade. 

Although our study has shown that LvIFI6-16 is an antiviral effector, which can regulate apoptosis-related genes and its self was activated by LvSTAT, there are many questions that remain to be further studied. For example, the transcription regulation mode of LvIFI6-16 is still unclear. We have predicted that there are many transcription factor binding sites on the LvIFI6-16 promoter (data not shown), which should be experimentally confirmed in the future. Lacking LvIFI6-16 antibodies, we neither conducted studies on the protein level of LvIFI6-16 interacting with its counterparts, nor explained how LvIFI6-16 regulates apoptosis-related genes. In conclusion, our study provides a solid foundation for studying the IFN antiviral pathway in invertebrates. 

## Figures and Tables

**Figure 1 viruses-14-01062-f001:**
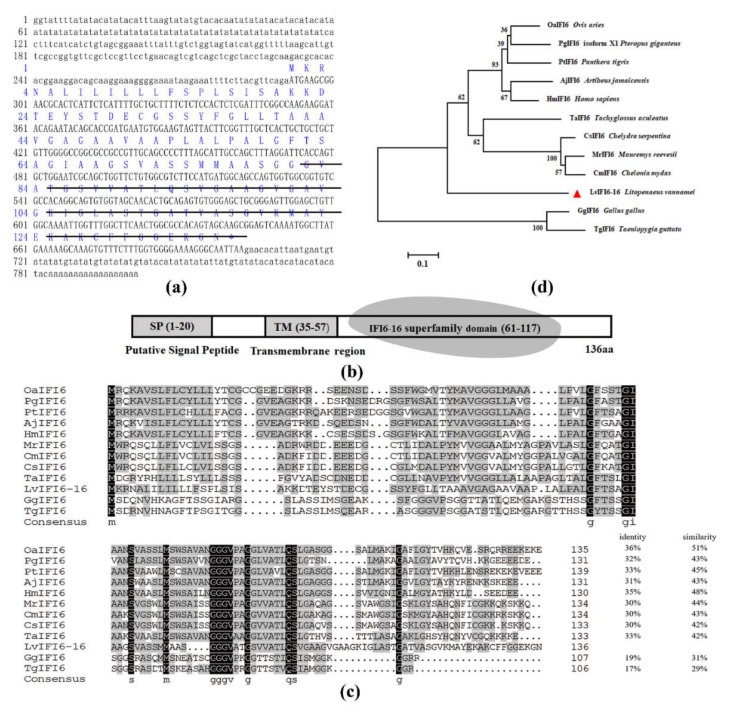
The sequence and phylogenetic analysis of LvIFI6-16. (**a**) The full-length cDNA sequence and deduced amino acid sequence of LvIFI6-16. The IFI-6-16 superfamily domain was underlined with a black line. (**b**) Structural features of LvIFI6-16 protein contained a putative signal peptide, a transmembrane region, and an IFI-6-16 superfamily domain. (**c**) Multiple sequence alignment of IFI6-16 homologs. The identical amino acid residues are shaded in black and the similar residues in gray. Amino acid identity and similarity of the LvIFI6-16 with other IFI6-16 homologs were shown on the right. (**d**) Evolutionary pattern of IFI6 homologs based on phylogeny. Phylogenetic tree analysis was based on the full-length amino acid sequences of IFI6-16 proteins (LvIFI6-16 was marked with a triangle) using MEGA7.1 software. Proteins analyzed listed below: *Litopenaeus vannamei* IFI6-16 (ON186541); *Panthera tigris* IFI6 (XP_042852994.1); *Gallus gallus* IFI6 (NP_001001296.1); *Pteropus giganteus* IFI6 (XP_039710223.1); *Mauremys reevesii* IFI6 (XP_039367217.1); *Tachyglossus aculeatus* IFI6 (XP_038614490.1); *Chelonia mydas* IFI6 (XP_007064928.2); *Artibeus jamaicensis* IFI6 (XP_037005569.1); *Taeniopygia guttata* IFI6 (NP_001184108.1); *Homo sapiens* IFI6 (BBG06130.1); *Ovis aries* IFI6 (XP_027821234.1); *Chelydra serpentina* IFI6 (KAG6932514.1).

**Figure 2 viruses-14-01062-f002:**
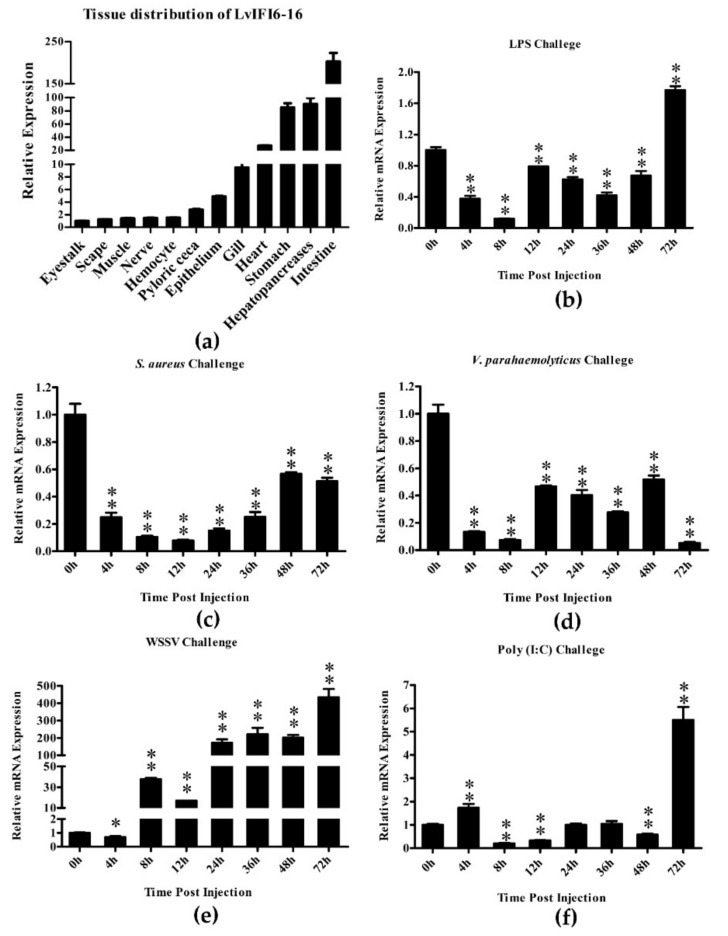
Tissue distribution of LvIFI6-16 in healthy *L. vannamei* and expression profiles of LvIFI6-16 in gills from pathogens or stimulants challenged *L. vannamei*. (**a**) Tissue distribution of LvIFI6-16 in healthy *L. vannamei*. Expression level in the eyestalk was used as control and set to 1.0. (**b**–**f**) Expression profiles of LvIFI6-16 in gills from LPS (**b**), *S. aureus* (**c**), *V. parahaemolyticus* (**d**), WSSV (**e**) and Poly (I:C) (**f**) challenged shrimps. Bars indicated the mean ± SD of three samples and statistical significances were calculated by the Student’s *t*-test (* *p* < 0.05, ** *p* < 0.01). Experiments were performed three times with similar results.

**Figure 3 viruses-14-01062-f003:**
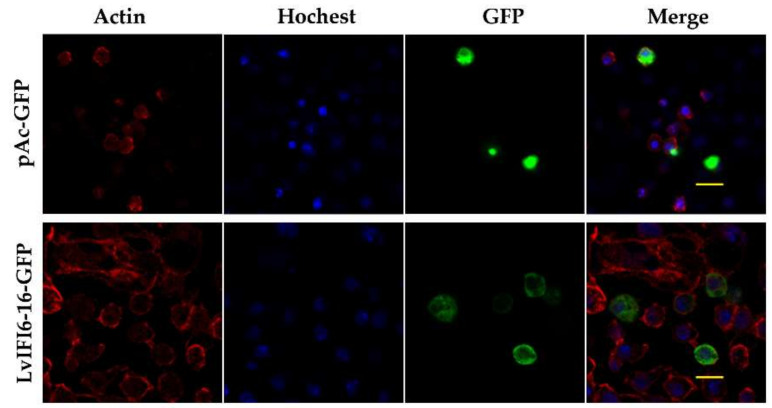
Subcellular localization of LvIFI6-16 in *Drosophila* S2 cells. Scale bar, 20 μm.

**Figure 4 viruses-14-01062-f004:**
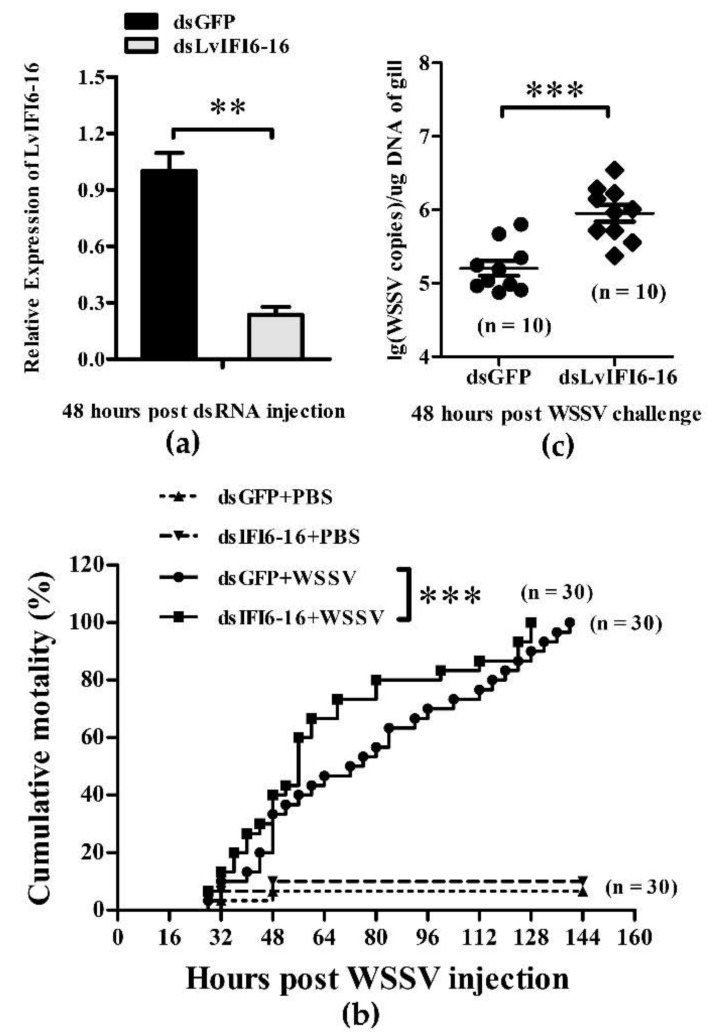
Functional analysis of LvIFI6-16 during WSSV infection. (**a**) Quantitative RT-PCR analysis of the silencing efficiencies of LvIF6-16. Bars indicated the mean ± SD of three samples and statistical significances were calculated by the Student’s *t*-test (** *p* < 0.01). (**b**) Cumulative mortality of LvIFI6-16-silenced shrimps after WSSV challenge. Differences in cumulative mortality levels between treatments were analyzed by log-rank (Mantel-Cox) test (*** *p* < 0.001). Experiments were performed three times with identical results. (**c**) WSSV genome copies in gill tissue (1 g) of LvIFI6-16 dsRNA and GFP dsRNA treated shrimps at 48 h post infection. Bars indicate the mean ± SD and statistical significances were calculated by the Student’s *t*-test (*** *p* < 0.001). Experiments were performed three times with similar results.

**Figure 5 viruses-14-01062-f005:**
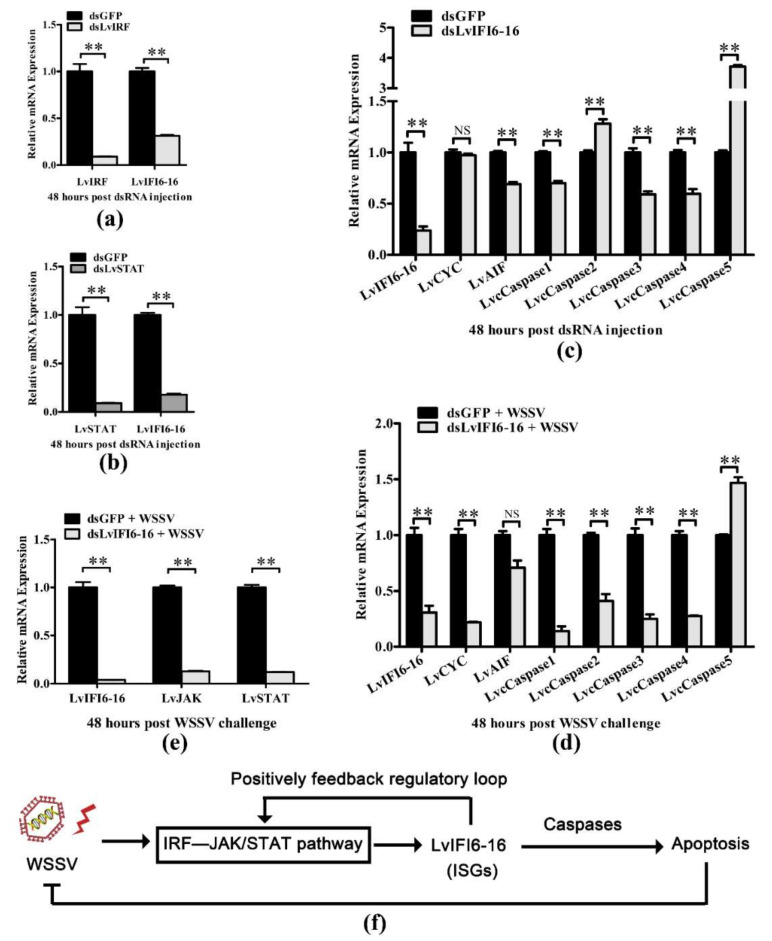
The regulatory effects between LvIFI6-16 and LvIRF, LvJAK, LvSTAT, or apoptosis-related genes before and after WSSV infection. (**a**) The transcription level of LvIFI6-16 in gills was regulated by LvIRF. (**b**) The transcription level of LvIFI6-16 in gills was regulated by LvSTAT. The internal control was LvEF-1α. Samples were taken at 48 h after dsLvIRF, dsLvSTAT, and dsGFP (as a control) injection. Bars indicated the mean ± SD of three samples and statistical significances were calculated by the Student’s *t*-test (** *p* < 0.01). (**c**) Transcription levels of apoptosis-related genes in gills of LvIFI6-16-knockdown shrimps. The internal control was LvEF-1α. Samples were taken at 48 h after injection with indicated dsRNA. Bars indicated the mean ± SD of three samples and statistical significances were calculated by the Student’s *t*-test (** *p* < 0.01; NS, not significant). (**d**) Transcription levels of apoptosis-related genes in gills of LvIFI6-16-knockdown shrimps after WSSV challenge. The internal control was LvEF-1α. Shrimps were injected with dsRNA of LvIFI6-16; after that, shrimp were infected with WSSV and samples were taken 48 h after WSSV infection. Bars indicated the mean ± SD of three samples and statistical significances were calculated by the Student’s *t*-test (** *p* < 0.01; NS, not significant). (**e**) LvIFI6-16 regulated the expression of LvJAK and LvSTAT in gills during WSSV infection. Samples were taken as described to (**d**). (**f**) A possible model of the LvIFI6-16 involving signaling regulation in response to WSSV infection. Pathogenic infection such as WSSV induced the activation of the IRF–JAK/STAT axis that triggered the expression of LvIFI6-16, which then resulted in the induction of several caspases that conferred antiviral apoptosis against WSSV. In addition, there could be a feedback regulatory loop mediated by LvIFI6-16 and JAK/STAT pathway. Bars indicated the mean ± SD of three samples and statistical significances were calculated by the Student’s *t*-test (** *p* < 0.01). Experiments were performed three times with similar results.

**Table 1 viruses-14-01062-t001:** Summary of primers in this study.

Name	Sequence (5’-3’)
**RACE**	
5’RACE-1	TGAGTGCGTTCCGCTTCAT
5’RACE-2	TCCTTGCTGTCCTTCCGTGTG
3’RACE-1	AGCTGCGGGAGTTGGAGCTGCT
3’RACE-2	AATTAAGAACACATTAATGAATGTA
**Quantitative RT-PCR**	
qPCR-LvEF-1α-F	TATGCTCCTTTTGGACGTTTTGC
qPCR-LvEF-1α-R	CCTTTTCTGCGGCCTTGGTAG
qPCR-LvAIF-F	TCTCTGGTGAGGGTGAAGCTCCCTA
qPCR-LvAIF-R	CTCCTTTCTTTCCCGTTCCATTGTT
qPCR-LvCYC-F	GAACGCGTCCCGGGGTTTC
qPCR-LvCYC-R	CTCGGTCTGCACATTCGGTCT
qPCR-Lvcasp1-F	CCGGGGCAAGAGGGCGGAGGAATAT
qPCR-Lvcasp1-R	CGGCACTGGGTCGCGGTTTGAGAGC
qPCR-Lvcasp2-F	ATGGCTCGTGGTTCATTCAG
qPCR-Lvcasp2-R	CATCAGGGTTGAGACAATACAGG
qPCR-Lvcasp3-F	AGTTAGTACAAACAGATTGGAGCG
qPCR-Lvcasp3-R	TTGTGGACAGACAGTATGAGGC
qPCR-Lvcasp4-F	CATGCTTGACATACCCGATG
qPCR-Lvcasp4-R	TGTCCGGCATTGTTGAGTAG
qPCR-Lvcasp5-F	GAAGGAGTGAAGCTAAACGAGAC
qPCR-Lvcasp5-R	CAGTAGACCAGCAGATAAGGAAGT
qPCR-LvIF6-16-F	ACTCATTCTCATTTTGCTGCTTTTCTC
qPCR-LvIF6-16-R	CCACATTCATCGGTGCTGTATTCT
qPCR-LvSTAT-F	CACAGAAGGTGTCAGGGCTATT
qPCR-LvSTAT-R	GATGCGCTGCTGAAGACTATTT
qPCR-LvJAK-F	TTTTGTAGGATGCTTGAATGGGTA
qPCR-LvJAK-R	GATAGAGAAGAGAAGGCGTTGAT
**Protein expression**	
LvIFI6-16-F	GGGGTACCATCAAAATGAAGCGGAACGCACTCATTC
LvIFI6-16-R	CGGAATTCATTGCCCTTTTCCCCACCAAAG
GFP-F	GGTTCGAAATCAAAATGGTGAGCAAGGGCGAGGAG
GFP-R	TTGTTTAAACTTACTTGTACAGCTCGTCCATGC
**dsRNA template amplification**	
GFP-T7-F	GGATCCTAATACGACTCACTATAGGGTGGTCCCAGTTCTTGTT
GFP-R	TTCTTTGGTTTGTCTCCC
GFP-F	GTGGTCCCAGTTCTTGTT
GFP-T7-R	GGATCCTAATACGACTCACTATAGGTTCTTTGGTTTGTCTCCC
LvIfi6-16-T7-F	GGATCCTAATACGACTCACTATAGGGAAGCGGAACGCACTCAT
LvIfi6-16-R	AATTGCCCTTTTCCCCAC
LvIfi6-16-F	GAAGCGGAACGCACTCAT
LvIfi6-16-T7-R	GGATCCTAATACGACTCACTATAGGAATTGCCCTTTTCCCCAC
LvSTAT-T7-F	GGATCCTAATACGACTCACTATAGGTCAGTATGCCCAGTCCTT
LvSTAT-R	CCTAACTCTTTCCGTCTCC
LvSTAT-F	TCAGTATGCCCAGTCCTT
LvSTAT-T7-R	GGATCCTAATACGACTCACTATAGGCCTAACTCTTTCCGTCTCC
LvIRF-T7-F	GGATCCTAATACGACTCACTATAGGGCCGCCATCTTTCACCAA
LvIRF-R	TGTCGTAGGAATGCGAGGAG
LvIRF-F	GCCGCCATCTTTCACCAA
LvIRF-T7-R	GGATCCTAATACGACTCACTATAGGTGTCGTAGGAATGCGAGGAG
**Absolute quantitative PCR**	
WSSV32678-F	TGTTTTCTGTATGTAATGCGTGTAGGT
WSSV32753-R	CCCACTCCATGGCCTTCA
TaqMan probe WSSV32706	CAAGTACCCAGGCCCAGTGTCATACGTT

## Data Availability

Not applicable.

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
