# Peer review of "Interferon-Induced Protein 6-16 (IFI6-16) from Litopenaeus vannamei Regulate Antiviral Immunity via Apoptosis-Related Genes"

_viruses, 2022, doi:10.3390/v14051062_

Round 1

Reviewer 1 Report

Authors have cloned a mammalian homolog of ISGs, i.e. IFI6-16 from  whiteleg shrimps and characterized its molecular properties as well as its signal regulation in defense against WSSV. Most of the results are straight forward, except for Figure 5 which involved signaling pathway.

General comments: The writing can be more concise.  Many redundant words can be deleted.  Present tense can be used if citing published facts, instead of recited the experimental conditions of the literature

The manuscript title: the title is too long and there is a misspelling which should be "Litopanneus". I suggest shorten the title to ".......regulate antiviral immunity via apoptosis-related genes".

line 15: give the common name of  (whiteleg shrimp) after species name.

line 16: replace "immune function" with "immunity".

line 42: delete "some".

line 48: delete "has been shown to".

line 50: replace "has been shown to" with "is".

line 61: delete "interesting and".  Spell what is "IRF" and "Vago"?

line 65: delete "many aspects including".

line 119, lines 319, lines 327-328: the  name should be italicized.

line 138: through out the manuscript authors seem to use the word "knockdown, knock down" and "silencing, silence" interexchangeably. Please confirm that if this is precise in the field of molecular biology.

Figure 1b: enlarge the schematic presentation of 1b.

Figure 1c: is not addressed in the text. I suggest insert a few words in section 3.2 before Figure 1d.  

line 188-189: animal species name should be italicized.

line 196: delete "analysis"

line 226-228: it can be deleted.  This is redundant with lines 232-234 and the materials and methods.

line 236: throughout the manuscript, use "qRT-PCR", except for its first appearance.

line 266: delete "be able to"

Figure 5: in (a): there are two "LvIFI6-16". please clarify. According to the experimental purpose, this figure is confusing and should be separated into 2 or 3 figures.  Readers can be confused unless you establish a "working  hypothesis" like exemplified in the attached file or you should construct a more correct working hypothesis.  Without a conceptual presentation, readers will not understand what are you discussing and authors cannot continue their works in a logical  or convincing manner.

discussion: lines 333-355: discuss based on your newly constructed working model.

lines 338-339: rephrase and avoid using authors names and et al.

Author Response

Reviewer 1

Authors have cloned a mammalian homolog of ISGs, i.e. IFI6-16 from  whiteleg shrimps and characterized its molecular properties as well as its signal regulation in defense against WSSV. Most of the results are straight forward, except for Figure 5 which involved signaling pathway.

General comments: The writing can be more concise.  Many redundant words can be deleted.  Present tense can be used if citing published facts, instead of recited the experimental conditions of the literature.

Reply: Thanks, we have carefully checked the writing and revised the whole manuscript.

  1. The manuscript title: the title is too long and there is a misspelling which should be "Litopanneus". I suggest shorten the title to ".......regulate antiviral immunity via apoptosis-related genes".
  2. line 15: give the common name of  (whiteleg shrimp) after species name.
  3. line 16: replace "immune function" with "immunity".
  4. line 42: delete "some".
  5. line 48: delete "has been shown to".
  6. line 50: replace "has been shown to" with "is".
  7. line 61: delete "interesting and".  Spell what is "IRF" and "Vago"?
  8. line 65: delete "many aspects including".
  9. line 119, lines 319, lines 327-328: the name should be italicized.
  10. line 138: through out the manuscript authors seem to use the word "knockdown, knock down" and "silencing, silence" interexchangeably. Please confirm that if this is precise in the field of molecular biology.
  11. Figure 1b: enlarge the schematic presentation of 1b.
  12. Figure 1c: is not addressed in the text. I suggest insert a few words in section 3.2 before Figure 1d.  
  13. line 188-189: animal species name should be italicized.
  14. line 196: delete "analysis"
  15. line 226-228: it can be deleted.  This is redundant with lines 232-234 and the materials and methods.
  16. line 236: throughout the manuscript, use "qRT-PCR", except for its first appearance.
  17. line 266: delete "be able to"
  18. discussion: lines 333-355: discuss based on your newly constructed working model.
  19. lines 338-339: rephrase and avoid using authors names and et al.

Reply (1-19): Thank you very much, as your suggestion, we have corrected these writings in revised manuscript.

Figure 5: in (a): there are two "LvIFI6-16". please clarify. According to the experimental purpose, this figure is confusing and should be separated into 2 or 3 figures.  Readers can be confused unless you establish a "working hypothesis" like exemplified in the attached file or you should construct a more correct working hypothesis. Without a conceptual presentation, readers will not understand what are you discussing and authors cannot continue their works in a logical or convincing manner.

Reply: Thanks. As your suggestion, this figure has been separated into 2 figures in the revised figure 5 (a and b).

Reviewer 2 Report

Although invertebrates do not have acquired immune systems, they have anti-viral immunity analogous to IFN system of vertebrates. However, functions of IFN and IGS in invertebrates are largely unknown. 
In this manuscript, the authors identified IFI6-16 gene from L. vannamei. They also revealed the expression of LvIFI6-16 in different tissues, and the changes of expression after stimulation. By using RNAi, they clarified the existence of IRF-JAK/STAT-IFI6-16-caspase regulator axis in shrimp. And, very interestingly, knocking down IFI16-6 made shrimps less resistant to WSSV infection, indicating that LvIFI6-16 is important for anti-viral immunity in shrimp. However, mechanisms of LvIFI6-16 in anti-viral immunity have not been elucidated in this paper, and complete understanding of LvIFI6-16 function requires further investigations.

I think sufficient information is not provided in this manuscript. The authors need to clarify the following points. 
1) Why was IFI6-16 analyzed?
It is not clear why the authors focus on IFI6-16 among many ISGs. 
2) Important information about cloning of full length IFI6-16 cDNA is missing.
In section 2.1, the methods to identify 5’ and 3’ ends of cDNA were described. However, the main part of cDNA could not be obtained by these methods. In addition, the source of RNA to synthesize cDNA should be shown. And what are the sequences of UPM and NUP for RACE? Or if a kit was used, what is the product name and supplier? The authors have to rewrite this section by including such kinds of information. Finally, the sequence should be submitted to database, and the accession number should be indicated.
3) Were the results in Figure 2 reproducible?
In Figure 2, were they the results of one experiment with triplicates? If so, experiments should be repeated. And the method of statistical analysis should be indicated.
4) Is IFI6-16 a secretory protein?
As shown in Figure 1 and in line 169, LvIFI6-16 has a transmembrane domain. However, in line 317, LvIFI6-16 was considered as secretory protein. How is this discrepancy explained?
5) How long did the knockdown effect of dsRNA continue?
In Figure 4 (a), decrease of LvIFI6-16 was analyzed at the time of WSSV infection. Then, mortality was monitored until 144 hours. Did the expression of LvIFI6-16 stay at low levels during this period?

The following are minor concerns.
6) In section 2.1, fixed cells were incubated with antibodies. What are the target molecules of antibodies? Clone names and suppliers should be clarified.
7) In line 117, shrimps were divided into five groups. However, there were six group, i.e., S. aureus, V. parahaemolyticus, WSSV, LPS, poly(I*:C), and negative control groups. Is it correct?
8) The methods to inject stimulators or dsRNA were not clear. Tools and inoculation sites should be described.

Author Response

Reviewer 2

Although invertebrates do not have acquired immune systems, they have anti-viral immunity analogous to IFN system of vertebrates. However, functions of IFN and IGS in invertebrates are largely unknown. 
In this manuscript, the authors identified IFI6-16 gene from L. vannamei. They also revealed the expression of LvIFI6-16 in different tissues, and the changes of expression after stimulation. By using RNAi, they clarified the existence of IRF-JAK/STAT-IFI6-16-caspase regulator axis in shrimp. And, very interestingly, knocking down IFI16-6 made shrimps less resistant to WSSV infection, indicating that LvIFI6-16 is important for anti-viral immunity in shrimp. However, mechanisms of LvIFI6-16 in anti-viral immunity have not been elucidated in this paper, and complete understanding of LvIFI6-16 function requires further investigations.

I think sufficient information is not provided in this manuscript. The authors need to clarify the following points. 
1) Why was IFI6-16 analyzed?
It is not clear why the authors focus on IFI6-16 among many ISGs. 

Reply: Thanks. Although there are many ISGs found in vertebrates, only a few ISGs (a total of three) have been found in shrimp L. vannamei, and the LvIFI6-16 is firstly identified in this study. Another two ISGs are still under study in our lab.

2) Important information about cloning of full length IFI6-16 cDNA is missing.
In section 2.1, the methods to identify 5’ and 3’ ends of cDNA were described. However, the main part of cDNA could not be obtained by these methods. In addition, the source of RNA to synthesize cDNA should be shown. And what are the sequences of UPM and NUP for RACE? Or if a kit was used, what is the product name and supplier? The authors have to rewrite this section by including such kinds of information. Finally, the sequence should be submitted to database, and the accession number should be indicated.

Reply: Thanks. We used the SMARTer PCR cDNA Synthesis Kit from Clontech to execute the RACE experiment (added in line 95-97 of the revised manuscript). The Genbank accession number of LvIFI6-16 has been added in line 189 of the revised manuscript.

3) Were the results in Figure 2 reproducible?
In Figure 2, were they the results of one experiment with triplicates? If so, experiments should be repeated. And the method of statistical analysis should be indicated.

Reply: Thanks. All the experiments in Figure 2 were performed three times with similar results. We have noted this in line 242 of the revised manuscript.

4) Is IFI6-16 a secretory protein?
As shown in Figure 1 and in line 169, LvIFI6-16 has a transmembrane domain. However, in line 317, LvIFI6-16 was considered as secretory protein. How is this discrepancy explained?

Reply: Thanks for your kind suggestion. We have rewritten this part of discussion (Line 342-347). As we reconsider the structure of LvIFI6-16 and the localization of IFI6-16 in other species, we suppose that LvIFI6-16 is more likely to be an intracellular transmembrane protein rather than a secretory protein.

5) How long did the knockdown effect of dsRNA continue?
In Figure 4 (a), decrease of LvIFI6-16 was analyzed at the time of WSSV infection. Then, mortality was monitored until 144 hours. Did the expression of LvIFI6-16 stay at low levels during this period?

Reply: Thanks. According to other studies on dsRNA treatment in L. vannamei, the knockdown effect can last about 7 d (168 h) (Developmental & Comparative Immunology 2014 42 (2), 174-185). Also, we checked the silence efficiency and found it still work at 96 hours post dsRNA-LvIFI6-16 injection, but we have no samples at other time points.

The following are minor concerns.
6) In section 2.1, fixed cells were incubated with antibodies. What are the target molecules of antibodies? Clone names and suppliers should be clarified.

Reply: Thanks. We have revised these sentences as your suggestion.

7) In line 117, shrimps were divided into five groups. However, there were six group, i.e., S. aureus, V. parahaemolyticus, WSSV, LPS, poly(I*:C), and negative control groups. Is it correct?

Reply: Thanks. There were six groups in this experiment including five pathogens or stimuli treated groups and the PBS treated group as a control.

8) The methods to inject stimulators or dsRNA were not clear. Tools and inoculation sites should be described.

Reply: Thanks. We have revised these sentences as your suggestion.

Reviewer 3 Report

In this study, the authors developed IFI6-16 in Shrimp Ltopenaeus vannamei and they evaluated the antiviral response by challnege with LPS or live microbes. Ina ddition, the autthors assessed the role of IFI6-16 on caspases genes.

I have some comments

1- Figure 2a:  The authors need to discuss why the distribution of Lv IFI6-16 is highly variable among different organs

2- Figure 2: Data is very confusing especially different time points: With S. aureus/ Vibrio parahemolyticus: there is down regualtion kinetic at deveral time points. Ont he opther hane , with WSSV challenge, there is an upregulation of IFI16 overtime. While with LPS and poly IC, there is a mrked upregulation especially at 72 hr.  This results should be discussed and decribed correctly.

3- Figure 3: I do not understand why the authors used eyestalk at timepoint of 0. Eyestalk showed the lowest level of IFI16 expression.

4- In vivo experiment is not convincing: did the authors collect tissues from the animals and did some histological/ IF for virological and IFI16 staining.

Author Response

Reviewer 3

In this study, the authors developed IFI6-16 in Shrimp Litopenaeus vannamei and they evaluated the antiviral response by challnege with LPS or live microbes. In addition, the autthors assessed the role of IFI6-16 on caspases genes.

I have some comments

1- Figure 2a:  The authors need to discuss why the distribution of Lv IFI6-16 is highly variable among different organs.

Reply: Thanks. As your suggestion, we have discussed in line 355-358 of the revised manuscript.

2- Figure 2: Data is very confusing especially different time points: With S. aureus/ Vibrio parahaemolyticus: there is down regulation kinetic at deveral time points. On the other hand, with WSSV challenge, there is an upregulation of IFI16 overtime. While with LPS and poly I:C, there is a marked upregulation especially at 72 hr. This result should be discussed and described correctly.

Reply: Thanks. LvIFI6-16 was upregulated or downregulated in response to these stimuli, which maybe attribute to that these stimuli are different. For example, V. parahaemolyticus is a Gram-negative bacterium, while S. aureus is a Gram-positive bacterium. WSSV represents a DNA virus, while poly I:C represents a RNA virus. Besides, as your suggestion, we have added some discussion in the revised manuscript (Line 359-364).

3- Figure 3: I do not understand why the authors used eyestalk at timepoint of 0. Eyestalk showed the lowest level of IFI16 expression.

Reply: Thanks. The expression level of LvIFI6-16 among different tissues is lowest in the eye stalk, which was set to 1.0 and the expression levels of other tissues were normalized to eye stalk such that making the histogram more intuitive.

4- In vivo experiment is not convincing: did the authors collect tissues from the animals and did some histological/ IF for virological and IFI16 staining.

Reply: Thanks. Because the lack of a specific antibody targeting LvIFI6-16, we cannot do histological/ IF for virological and IFI16 staining. We will prepare a specific antibody of LvIFI6-16 for further research in the future.

Round 2

Reviewer 1 Report

I do not see much conceptual improvement  in Figure 5.  Perhaps  authors misunderstood what I meant.

In Figure 5,  you have 5 subfigures.  If you have done all 5 experiments  at the same time, then you will not see the logics and mutual relationships among the 5 subfigures.  What I want to see is (assuming you design experiment following the "known pathway" that you mentioned in the manuscript as your working model) you've designed and done a experiment (5a), you get  results (called Figure 5a) and interpreted the results. You think results 5a is consistent with or contradict to the known pathway (your current working model). Then  you designed another experiment (5b) to follow-up and get results (Figure 5b) and it is consistent with or contradict to the model. Then you decide experiment 5c and so on.  If you follow this step, I can see the logics flowing from 5a to 5b to 5c to 5d to 5e. If you have the logics, section 3.6 should be rewritten according to this logics. And I will clearly see your working model (may be similar or different from the known pathway).  Your any future experiments will be to examine this (your own) working model and refine your model in shrimps.

In this R1 version , the change you made is just cosmetics, and I do not see much mutual relationships between Figure 5a through 5e.  You simply laid out the results.

If you decide to resubmit, please provide a cleaner copy and marked the change in color.

Author Response

In Figure 5, you have 5 subfigures.  If you have done all 5 experiments at the same time, then you will not see the logics and mutual relationships among the 5 subfigures.  What I want to see is (assuming you design experiment following the "known pathway" that you mentioned in the manuscript as your working model) you've designed and done an experiment (5a), you get results (called Figure 5a) and interpreted the results. You think results 5a is consistent with or contradict to the known pathway (your current working model). Then, you designed another experiment (5b) to follow-up and get results (Figure 5b) and it is consistent with or contradict to the model. Then you decide experiment 5c and so on.  If you follow this step, I can see the logics flowing from 5a to 5b to 5c to 5d to 5e. If you have the logics, section 3.6 should be rewritten according to this logic. And I will clearly see your working model (may be similar or different from the known pathway).  Your any future experiments will be to examine this (your own) working model and refine your model in shrimps.

In this R1 version, the change you made is just cosmetics, and I do not see much mutual relationships between Figure 5a through 5e.  You simply laid out the results.

If you decide to resubmit, please provide a cleaner copy and marked the change in color.

Reply: Thank you for your good suggestion. We reorganized section 3.6, and added a relationship summary figure 5f according to your suggestion in the R1 version. (Figure 5f). A possible model of the LvIFI6-16 involving signaling regulation in response to WSSV infection. Pathogenic infection such as WSSV induced the activation of IRF-JAK/STAT axis that triggered the expression of LvIFI6-16, which then resulted in the induction of several caspases that conferred antiviral apoptosis against WSSV. Also, there could be a feedback regulatory loop mediated by LvIFI6-16 and JAK/STAT.

Reviewer 2 Report

I found that most of my concerns were addressed in the revised manuscript. However, I still cannot understand how “full length” sequence was obtained. 5’ RACE1 primer hybridizes to nucleotide position from 310 to 292, while 3’ RACE1 primer hybridizes from 579 to 595. How did you get the sequence between 331 and 578?

Author Response

I found that most of my concerns were addressed in the revised manuscript. However, I still cannot understand how “full length” sequence was obtained. 5’ RACE1 primer hybridizes to nucleotide position from 310 to 292, while 3’ RACE1 primer hybridizes from 579 to 595. How did you get the sequence between 331 and 578?

Reply: Thank you! Based on L. vannamei EST in our Lab (Li et al. PLoS One 2012, 7 (10), e47442), an EST that contains the sequence between 331 and 578 and encodes part amino acids of the L. vannamei IFI6-16 protein was used to design the 5’ RACE1 and 3’ RACE1 primers (described in method).  

Reviewer 3 Report

no further comments

Author Response

Thank you!